# Obstructive Sleep Apnea in Heart Failure: Current Knowledge and Future Directions

**DOI:** 10.3390/jcm11123458

**Published:** 2022-06-16

**Authors:** Shahrokh Javaheri, Sogol Javaheri

**Affiliations:** 1Division of Pulmonary and Sleep Medicine, Bethesda North Hospital, Cincinnati, OH 45242, USA; shahrokhjavaheri@icloud.com; 2Division of Cardiology, The Ohio State University, Columbus, OH 43210, USA; 3Division of Pulmonary and Sleep Medicine, University of Cincinnati, Cincinnati, OH 45242, USA; 4Division of Sleep and Circadian Disorders, Brigham and Women’s Hospital, Harvard Medical School, Boston, MA 02130, USA

**Keywords:** obstructive sleep apnea, heart failure, continuous positive airway pressure

## Abstract

Obstructive sleep apnea (OSA) is highly prevalent among patients with asymptomatic left ventricular systolic and diastolic dysfunction and congestive heart failure, and if untreated may contribute to the clinical progression of heart failure (HF). Given the health and economic burden of HF, identifying potential modifiable risk factors such as OSA and whether appropriate treatment improves outcomes is of critical importance. Identifying the subgroups of patients with OSA and HF who would benefit most from OSA treatment is another important point. This focused review surveys current knowledge of OSA and HF in order to provide: (1) a better understanding of the pathophysiologic mechanisms that may increase morbidity among individuals with HF and comorbid OSA, (2) a summary of current observational data and small randomized trials, (3) an understanding of the limitations of current larger randomized controlled trials, and (4) future needs to more accurately determine the efficacy of OSA treatment among individuals with HF.

## 1. Introduction

According to the American Heart Association 2022 Statistics [1], over 8 million people 18 years or older will have heart failure (HF) in 2030. Despite treatment advances, the prevalence, burden, and costs of HF continue to increase. The healthcare costs associated with HF exceed 30 billion dollars annually and over 50% of these costs are associated with hospitalizations [1]. Obstructive sleep apnea (OSA) is highly prevalent in, and may contribute to the progression of, both HF with reduced ejection fraction (HFrEF) and HF with preserved ejection fraction (HFpEF), potentially reflecting an important modifiable risk factor. The prevalence of OSA ranges from 20% to up to 60% among the HF population, with rates of OSA typically running higher among those with HFpEF as compared with HFrEF [2,3,4,5,6]. Observational data has shown that OSA is independently associated with poor quality of life, excess rehospitalization, and premature mortality among patients with HF [5]. Notably, multiple observational studies have demonstrated that effective treatment of OSA may decrease hospital readmission rates and improve survival [7,8,9]. To date, there are no randomized controlled trials assessing continuous positive airway pressure (CPAP) therapy in an HF with comorbid OSA population. Prior RCTs using CPAP to treat OSA have shown no benefit on secondary prevention of cardiovascular diseases but have shown improvement in quality-of-life measures. The purpose of this review is to review pathophysiologic mechanisms underlying the association between OSA and HF, to summarize current observational and randomized data, and to characterize the need for trials that can address important questions in real world patients and in specific subgroups of HF patients who may be more likely to benefit from OSA treatment.

## 2. OSA and HF Pathophysiology

OSA is characterized by repetitive upper airway closure due to soft tissue collapse and genioglossus muscle relaxation in the upper airway resulting in apneas (cessation of breathing for 10 s or longer) and hypopneas (reductions in breathing coupled with desaturation and/or arousal). Obesity and rostral fluid shifts both contribute to upper airway narrowing and collapse. Like the general population, obese individuals with HF are also more prone to upper airway closure related to fat deposition in the upper body, including visceral fat and tongue and throat fat [10]. At the same time, fluid retention and edema, particularly during HF decompensation, can contribute to upper airway closure in the supine position. Translocation of fluid from the lower extremities to the neck may cause vascular congestion and edema of the pharyngeal area [11]. Regardless of the contributing factors, the downstream effects include intermittent hypoxia and hyper-hypocapnia, repetitive arousals, and large negative intrathoracic pressure swings. Hypoxia, hypercapnia, and arousals lead to autonomic dysregulation with heightened sympathetic activity and reduce parasympathetic tone as well as hypothalamic pituitary axis dysregulation. Intermittent hypoxia reoxygenation leads to production of oxygen free radicals, oxidative stress, and upregulation of inflammatory cascades such as NF kb and TNF-alpha. Finally, negative intrathoracic pressure swings may result in increased atrial stretch (facilitating atrial fibrillation), left ventricular transmural pressure and afterload, and myocardial oxygen demand [5,11].

## 3. Epidemiology

Multiple observational studies suggest that OSA is independently associated with excess hospital readmission [9,10] and that treatment may lower the rate of readmissions [7,8,9]. Specifically, severe OSA has been independently associated with 1.5 times higher readmission of HF patients when compared with those without OSA [9]. Observational studies also suggest OSA is independently associated with premature mortality in individuals with comorbid HF [7,8,9,12,13] and that treatment of OSA attenuates this risk [7,9,12,13]. In the largest study, among 30,000 Medicare beneficiaries newly diagnosed with HF, the treatment of SDB was associated with decreased readmission, health care cost, and mortality [7]. Two other studies have shown that effective treatment of OSA with CPAP improves survival in patients with comorbid HF, particularly in those who are compliant with CPAP [9,12]. Long-term randomized control trials (RCTs) in this population are not available, but there is a critical need to assess how effective treatment of OSA affects the clinical course of HF and hard outcomes.

## 4. Acute Decompensated Heart Failure

Multiple observational studies have shown a high prevalence of SDB, particularly OSA in patients admitted to the hospital for HF decompensation. In a multi-center study from Brazil, consecutive patients with confirmed acute cardiogenic pulmonary edema (ACPE) underwent polygraphy following clinical stabilization [14]. Approximately 100 patients were included in the final analysis, of whom 79 had HFpEF with LVEF greater than or equal to 50%. A total of 61% of the patients had OSA defined as an apnea-hypopnea index (AHI) greater than or equal to 15 events/h based on polygraphy. The mean follow-up was 1 year and the primary outcome was ACPE recurrence. Higher incident rates of ACPE recurrence (25 vs. 6 episodes; *p* = 0.01) and myocardial infarction (15 vs. 0 episodes; *p* = 0.0004) were observed in patients with OSA compared with those without OSA. All 17 deaths occurred in the OSA group (*p* = 0.0001). In a Cox proportional hazards regression analysis, OSA was independently associated with ACPE recurrence (hazard ratio (HR), 3.3 [95% CI, 1.2–8.8], *p* = 0.01), incidence of myocardial infarction (HR, 2.3 [95% CI, 1.1–9.5]; *p* = 0.02), cardiovascular death (HR, 5.4 [95% CI, 1.4–48.4]; *p* = 0.004), and total death (HR, 6.5 [95% CI, 1.2–64.0]; *p* = 0.005). Among the patients with OSA who presented with ACPE recurrence or who died, AHI and hypoxemic burden and rates of sleep-onset ACPE were significantly higher [14].

Given the high prevalence of OSA in HFpEF and HFrEF and supportive observational data, OSA may represent a modifiable risk factor. This is particularly important as HFpEF remains highly prevalent and thus far pharmacological trials have not shown a drug therapy that could improve survival as the primary outcome, though a recent sodium–glucose co-transporter 2 (SGLT-2) inhibitor trial has demonstrated improved survival in the composite endpoint of hospital admission and mortality [15].

One RCT in acute decompensated HF randomized 150 patients with HFrEF who were diagnosed with OSA during hospitalization to a CPAP therapy arm (*n* = 75) or control arm (*n* = 75). All participants received guideline-directed therapy for HF decompensation. Exploratory analysis revealed that 6 months after discharge, there was over a 60% decrease in readmissions for patients who used PAP > 3 h/night compared with those who used PAP < 3 h/night (*p* < 0.02) and compared with controls (*p* < 0.04) [13]^.^

## 5. Limitations of Randomized Control Trials for OSA Treatment in HF

There have been multiple RCTs, primarily in participants without HF, assessing composite cardiovascular endpoints. Notably, all the trials enrolled individuals with established CVD and/or cerebrovascular disease. The largest OSA RCT to date, the SAVE trial, randomized 2717 patients with established cardiovascular disease (a minority with HF), at least moderate or severe OSA to CPAP, plus usual care versus usual care alone for almost four years. CPAP use did not reduce the composite cardiovascular endpoint, though in secondary analyses there was a lower risk of cerebrovascular events among patients using CPAP for at least 4 h per night [16]. Limitations to this trial (as well as other OSA RCTs) include low adherence to CPAP, minimally symptomatic population selection (largely excluding patients with very severe OSA and/or hypoxia), and reduced generalizability. Regarding adherence, various studies have suggested a linear relationship between hours of CPAP use and change in blood pressure. A metanalysis estimated a 1.39 mm Hg decrease in 24-h mean blood pressure for each 1-h increase in effective nightly use of CPAP [17]. It is feasible that below a certain threshold, too few hours of CPAP use may not confer cardiovascular and metabolic benefit. Additionally, there is data suggesting that sleepier patients [18] and those with more severe hypoxemic burden [19,20] have higher incident cardiovascular risk and therefore might benefit the most with CPAP therapy. However, in most trials, patients with more severe apnea and hypoxemic burden and those with excessive sleepiness are excluded, including the SAVE trial. In the SAVE trial, patients with ESS >15 were excluded and the average ESS was approximately 7 (under 10 is considered within normal limits), suggesting that most patients were not sleepy [16]. Since less symptomatic patients are less likely to benefit from CPAP therapy, this could also contribute to lower CPAP adherence. Overall, participants enrolled had established CVD, did not have severe hypoxic burden, and were generally non-sleepy; the cardinal symptom of OSA for which many patients seek treatment. Yet, observational studies, including the Wisconsin Sleep Cohort Study [21], the Busselton Health Study [22], and the Sleep Heart Health Study [23], revealed that only severe OSA was associated with premature mortality. Similarly, recent studies have suggested that hypoxemic burden is an important predictor of mortality and adverse cardiovascular sequelae [19,20]. Therefore, the patient populations studied in the SAVE trial (as with other RCTs to date) do not reflect the population of patients treated clinically.

## 6. OSA Phenotyping for Targeting Personalized Therapy

OSA is a heterogeneous disease with distinct endo/phenotypes. Individuals have different symptoms, clinical presentations, and risk factors and may respond differently to the same therapy. Determining which subgroup of patients (whether using symptoms such as sleepiness, PSG measures such as hypoxic burden, or other subtypes such as the insomnia and OSA or COMISA subtype) may have higher CVD risk and respond best to treatment may be a critical point [20,24]. In the Icelandic study, three phenotypes were identified: (1) the disturbed sleep group who were the OSA patients most likely to suffer from insomnia symptoms (and mean ESS < 10), (2) minimally symptomatic OSA patients (most normal ESS scores), and (3) excess daytime sleepiness (mean ESS 15.7 ± 0.6) patients. Differences among the three groups were not explained by obesity, age, sex, or AHI severity [25]. Prior studies have shown that the excessively sleepy subtype may be more strongly associated with a prevalence of HF and may also have a higher degree of CVD risk [26], potentially representing a symptomatic biomarker for CVD risk. Unfortunately, however, these are the very patients who are excluded from RCTs for lack of feasibility and ethical concerns. These patients may be at increased risk of motor vehicle collisions due to drowsy driving and may refuse to participate in a study where they will be in a control arm without effective long-term treatment. In order to effectively study cardiovascular outcomes and mortality, long-term follow up is required, which is ethically complex.

## 7. Pitfall of the AHI and Alternate Metrics

It is not yet clear what the best metric for measuring OSA disease burden is in HF patients. The AHI does not capture the depth or duration of upper airway obstruction, hypoxic burden, or REM versus NREM preponderance of respiratory events. There may be other metrics to quantify the severity of sleep apnea in the HF population, including measures of hypoxia (i.e., time spent < 90% oxygen saturation, hypoxic burden, measured as the area under the desaturation curve from respiratory event and pre-event baseline). Arousal burden (number and intensity of arousals), REM versus NREM AHI, duration of respiratory events (in addition to frequency), changes in heart rate in response to arousals, or other biomarkers may also present potential metrics of interest. Studies are needed to investigate metrics that may allow for better phenotyping and patient selection first in the general population and then in the HF population. The hypoxic burden and pulse rate response to respiratory events or arousal are currently under investigation [27], however more data on reliability and ease of measurement are needed before clinical application.

## 8. Other Therapeutic Options

It is widely known that CPAP can be difficult to tolerate, particularly in the HF population. Alternate treatments include oral appliances, hypoglossal nerve stimulation, and positional therapy in patients with supine-dominant OSA. In CPAP-intolerant individuals, custom-made oral appliances and hypoglossal nerve stimulation are recommended, though limited studies are available in HF. Studying the efficacy of alternate therapeutic options, including oral appliance therapy and hypoglossal nerve stimulation, will also be important in the HF population.

## 9. Summary and Remaining Issues

OSA is highly prevalent in all types of HF. It is biologically plausible that OSA leads to adverse cardiovascular sequelae, and long term pathobiologic consequences of sleep apnea may include sustained increases in sympathetic activity, endothelial dysfunction, oxidate stress, and up-regulation of inflammatory cytokines, ultimately leading to a variety of cerebrocardiovascular complications. Observational studies have demonstrated associations between OSA with excess rehospitalization and mortality among HF patients, and multiple observational studies have demonstrated that effective treatment of OSA decreases hospital readmission and improves survival [10,12,28]. Small prospective [29] and randomized trials [13,30] also show improvement in intermediate outcomes with use of PAP therapy, including reductions in blood pressure and ejection fraction with effective treatment of OSA. However, larger RCTs on composite outcomes have been negative or inconclusive, though they do consistently demonstrate improved quality of life and mood with use of PAP. To date, there are no randomized controlled trials assessing continuous positive airway pressure (CPAP) therapy in an HF with comorbid OSA population, and future studies with improved design and implementation and using alternative therapeutic options, not only PAP devices, are needed to determine whether OSA treatment improves morbidity and mortality in HF patients beyond quality-of-life measures. Additionally, studies are needed to better characterize phenotypes of OSA and objective measures to help determine who may respond best to CPAP (i.e., sleepy versus non-sleepy patients). Use of other metrics beyond the AHI and inclusion of specific OSA phenotypes may allow for more targeted patient selection in future trials. Until then, treatment options should start with evidence-based therapy for HF, and treatment for OSA in HF should be targeted to improve AHI and alleviate patient symptoms with the goal of improving mood, quality of life, and blood pressure. More targeted and thoughtful selection of study populations, larger populations, and higher adherence to PAP are all challenges that must be overcome in OSA RCTs in order to adequately address the pressing questions of whether OSA is indeed a modifiable risk factor for HF and whether CPAP can improve survival and other cardiovascular endpoints.

## Data Availability

Not applicable.

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
