# Peer review of "Obstructive Sleep Apnea in Heart Failure: Current Knowledge and Future Directions"

_jcm, 2022, doi:10.3390/jcm11123458_

Round 1

Reviewer 1 Report

Javaheri S et al summarized the association between obstructive sleep apnea (OSA) and heart failure (HF). The authors explain that both HF and OSA share some clinical background and pathophysiology.

As for treatment , the authors introduced the efficacy of CPAP therapy and mentioned the limitation of past RCTs. One of the reasons why no effective treatment evidence exist is that both HF and OSA have a variety of clinical features. Thus, we need to treat these patients with knowledge of their clinical phenotyping. Finally, the authors suggested the future proposal for treatment of OSA with HF.

I would like to ask you make figures which explain the pathophysiological association between OSA and HF, and to make table or figure which introduce specific phenotypes of HF with OSA.

I think this review is clear and legible.

I do not haven any concerns about this review.

Author Response

Thank you for your comments.  We have now attached two figures which we hope enhances the content of the review.  

Reviewer 2 Report

This could be potentially interesting paper, however, the presentation format needs to be improved following these suggestions:

  • The paper title is too generic, which does not reflect more specific focus described in Abstract.
  • Abstract could be better organized i.e., please identify the objectives of the work and of the paper. For example, the paper could be state of the art survey of our current knowledge about OSA.
  • The initial paragraph below keywords is not part of any section.
  • There needs to be some explanation at the beginning about the objectives of the paper, outlining the paper contributions as well as explaining the paper structure.
  • Some sentences are grammatically incorrect or more precisely incomplete. 
  • In the description, it is often unclear whether the statements are taken from the literature, or the authors refer to their own experiments.
  • The purpose of Summary section at the end of the paper is unclear. It would be useful to see conclusions and discussion of the main findings of the paper.
  • Other issues: please do not use colon at the end of sub-section titles, use the unified capitalization of section/sub-section titles, replace >=18 symbol with words, the term 'HF population' is a bit vague, it is unclear why 'r' in HFrEF is not capitalized

Round 2

Reviewer 2 Report

Please consider the following further revisions:

- Paper title: remove 'a summary of'

- Abstract: better to use 'focused review' or similar than 'narrative review'; Please add a sentence or two before listing contributions to justify your review, e.g. what can be said about the state of current literature on this topic?

- Section/subsection titles do not have uniform capitalization of words, what are not they numbered?

- I cannot see a paragraph in Introduction explaining the structure of the remaining paper, and explaining the strategy how the review has been conducted.

- Section title: 'Pitfalls of AHI and (of) other metrics'

- Section title: 'Summary and remaining issues' or something similar would be better than 'future needs'

Author Response

Thank you for the additional comments which we hope have further enhanced this review.

C1. Paper title: remove 'a summary of'

R1. We have removed "a summary of" in the title.

C2. Abstract: better to use 'focused review' or similar than 'narrative review'; Please add a sentence or two before listing contributions to justify your review, e.g. what can be said about the state of current literature on this topic?

R2.  We have changed narrative review to focused review.  We have added the following sentences to the abstract: "Given the health and economic burden of HF, identifying potential modifiable risk factors such as OSA and whether appropriate treatment improves outcomes is of critical importance.  Identifying the subgroups of patients with OSA and HF who would benefit most from OSA treatment is another important point."

C3. Section/subsection titles do not have uniform capitalization of words, what are not they numbered?

R3. Thank you for highlighting this oversight. We have corrected this so capitalization is now uniform.

C4.  I cannot see a paragraph in Introduction explaining the structure of the remaining paper, and explaining the strategy how the review has been conducted.

R4.  This last paragraph of the introduction summarizes the body of the paper and it reads: 

"The purpose of this review is to review pathophysiologic mechanisms underlying the association between OSA and HF, to summarize current observational and randomized data, and to characterize the need for trials that can address important questions in real world patients and in specific subgroups of HF patients who may be more likely to benefit from OSA treatment."  We feel that the ensuing sections of the paper cover these topics in that order and so did not add any additional comments on paper structure.    

R.5 Section title: 'Pitfalls of AHI and (of) other metrics'

C5. We changed the section title to read "Pitfalls of AHI and Alternate Metrics" because we are not describing the pitfalls of the other metrics but introducing them as potential alternatives or additions to the AHI.

R6. Section title: 'Summary and remaining issues' or something similar would be better than 'future needs'

C6. We have changed this title to "Summary and Remaining Issues."
